# Candida Chorioamnionitis in Mothers with Gestational Diabetes Mellitus: A Report of Two Cases

**DOI:** 10.3390/ijerph18147450

**Published:** 2021-07-13

**Authors:** Elia Shazniza Shaaya, Siti Atiqah Abdul Halim, Ka Wen Leong, Kevin Boon Ping Ku, Pei Shan Lim, Geok Chin Tan, Yin Ping Wong

**Affiliations:** 1Department of Pathology, Faculty of Medicine, Universiti Kebangsaan Malaysia, Kuala Lumpur 56000, Malaysia; elia_shaz@yahoo.com (E.S.S.); teeqah@hotmail.co.uk (S.A.A.H.); kawen_leong@yahoo.com (K.W.L.); kevinku128@gmail.com (K.B.P.K.); 2Department of Obstetrics & Gynaecology, Faculty of Medicine, Universiti Kebangsaan Malaysia, Kuala Lumpur 56000, Malaysia; pslim@ppukm.ukm.edu.my

**Keywords:** placenta, chorioamnionitis, *Candida* spp., umbilical cord, diabetes mellitus

## Abstract

*Background:**Candida* chorioamnionitis is rarely encountered, even though vulvovaginal candidiasis incidence is about 15%. Interestingly, it has characteristic gross and histological findings on the umbilical cord that are not to be missed. *Case Report:* We report two cases of *Candida* chorioamnionitis with presence of multiple yellowish and red spots of the surface of the umbilical cord. Microscopically, these consist of microabscesses with evidence of fungal yeasts and pseudohyphae. The yeasts and pseudohyphae were highlighted by periodic acid– Schiff and Grocott methenamine silver histochemical stains. Both cases were associated with a history of gestational diabetes mellitus. *Discussion:* Peripheral funisitis is a characteristic feature of *Candida* chorioamnionitis. It is associated with high risk of adverse perinatal and neonatal outcomes, such as preterm delivery, stillbirth and neonatal death. We recommend careful examination of the umbilical cord of mothers with gestational diabetes mellitus.

## 1. Introduction

*Candida* chorioamnionitis with fungal invasion of the umbilical cord was first described by Benirschke et al. in 1958 [1]. It is rarely encountered but may cause neonatal morbidity and mortality such as premature birth, perinatal death and congenital infection [2]. It has typical morphological features in the umbilical cord. However, due to its rarity, this may be easily overlooked and missed during histological examination. *Candida albicans* is the most encountered fungal organism causing acute chorioamnionitis and funisitis. Here, we present two cases of *Candida* chorioamnionitis in pregnancy with gestational diabetes mellitus. Consequently, one of the infants developed jaundice and was treated with phototherapy.

## 2. Case Report

### 2.1. Case 1

A 34-year-old, gravida 1, para 0 + 1 woman presented to a tertiary hospital with 2 h of contraction pain at 35 weeks of gestation. She was afebrile and her vital signs were stable. She had a history of miscarriage at 17 weeks of gestation in her first pregnancy. Antenatally, she was found to have gestational diabetes mellitus and was on diet control. Her blood glucose level fluctuated throughout the pregnancy with pre-meal glucose of 4.6–4.7 mmol/L and 2 h post-meal glucose of ranged between 5.1 to 16.8 mmol/L. Her HbA1c was 5.4%. At 29 weeks of gestation, a high vaginal swab culture was positive for *Candida* spp. and was treated with Clotrimazole pessary. There was no history of amniocentesis procedure performed during the pregnancy. The foetal anomaly scan showed a normal well-developed foetus.

Laboratory investigation showed a raised white cell count (17.5 × 10^9^/L) with predominantly neutrophils (14.1 × 10^9^/L). Artificial rupture of membrane was performed and showed clear amniotic fluid. A male infant weighing 2.59 kg was delivered via spontaneous vaginal delivery, with Apgar scores of 8 and 10 at 1 and 5 min, respectively. The infant was admitted to the neonatal ward for observation with presumed sepsis and unexplained prematurity.

Placenta and umbilical cord were sent for histopathological examination. The placenta weighed 500 g. Grossly, the amniotic membrane appeared opaque and the umbilical cord demonstrated yellow and haemorrhagic spots on the surface (Figure 1). Serial sectioning of the placenta showed unremarkable cut surfaces. Microscopic examination of the placenta showed dense neutrophilic infiltrates with karyorrhectic debris in the foetal membrane and decidua, with presence of foetal vasculitis. Examination of umbilical cord revealed multiple microabscesses and fungal bodies at the periphery (Figure 2A,B), along with umbilical phlebitis and arteritis. Periodic acid–Schiff (PAS) and Grocott methenamine silver (GMS) histochemical stains highlighted the fungal bodies at the periphery of the umbilical cord (Figure 2C,D). A final diagnosis of acute necrotising chorioamnionitis with umbilical peripheral funisitis, phlebitis and arteritis, associated with *Candida* infection was made. A repeat high vaginal swab culture was performed on the mother, and it was positive for *Candida* spp. and Group B *Streptococcus*.

The infant was given intravenous penicillin and gentamicin. He developed neonatal jaundice at 36 h of life and required intensive phototherapy. He was discharged at day 4 of life after a reduction in total serum bilirubin and completion of intravenous antibiotics. However, he was re-admitted at day 8 due to unresolved neonatal jaundice and was treated with phototherapy. The jaundice was completely resolved on day 9 of life.

### 2.2. Case 2

A 28-year-old, gravida 1, para 0 woman, at 39 weeks and 3 days of gestation presented to our hospital with leaking liquor for more than 18 h and contraction pain. She was afebrile at the time of presentation. Antenatally, she was diagnosed with gestational diabetes mellitus at 21 weeks of gestation after a modified glucose tolerance test (MGTT) showing a fasting blood glucose of 5.2 mmol/L and 2 h post-prandial glucose of 7.4 mmol/L. She was on diet control. Her blood glucose level was fairly well-controlled throughout the pregnancy with pre-breakfast glucose of 4.4–5.5 mmol/L and 2 h post-meal glucose of between 5.0 to 6.3 mmol/L. She also had mild anaemia with haemoglobin levels between 10.9 and 12.6 gm/dL.

Placenta and umbilical cord were submitted for histopathological examination. The placenta weighed 600 g. Grossly, the amniotic membrane appeared opaque. Serial sections of the placenta showed unremarkable cut surfaces. Microscopically, there was dense neutrophilic infiltration in the foetal membrane with neutrophilic karyorrhexis. Foetal chorionic vasculitis was identified. The umbilical cord demonstrated umbilical phlebitis and arteritis and multiple foci of peripheral microabscesses. PAS and GMS histochemical stains revealed the presence of yeasts and pseudo-hyphae at the periphery of the umbilical cord. A diagnosis of acute necrotising chorioamnionitis with umbilical peripheral funisitis, phlebitis and arteritis, associated with *Candida* infection was made. Placental swab culture study showed *Candida* spp.

During labour, she had an episode of temperature spike of 37.9 °C and was given intravenous ampicillin. The amniotic fluid was lightly meconium-stained and non-foul smelling. Her white cell count and c-reactive protein (CRP) were raised, 16.3 × 10^9^/L and 5.52 mg/dL, respectively. Her urine culture showed no growth. She delivered a baby boy with birth weight of 3.15 kg and Apgar scores of 9 in 1 min and 10 in 5 min. The infant was admitted for paediatric care due to presumed sepsis. At 8 h of life, his oxygen saturation dropped to 85% SpO_2_ under room air, and was admitted to neonatal intensive care unit for continuous positive airway pressure. He received IV gentamicin and penicillin for 5 days. He was started on oral nystatin after both the placental examination and swab culture demonstrated *Candida* infection. He was discharged well after completion of antibiotic and antifungal medications.

## 3. Discussion

*Candida* spp., particularly *Candida albicans,* inhabits the vaginal mucosa of otherwise healthy individuals as harmless commensal, as with some bacteria [3]. This yeast is regarded as opportunistic pathogen that takes advantage of its host following perturbation of local microbiota, a breach in mucosal barrier or compromised immune defence. Vaginal microecological environment is a unique and dynamic ecosystem, in which its functional equilibrium can be affected by a wide variety of factors. An elevated level of oestrogen during pregnancy may cause the vagina to produce more glycogen, facilitating the colonisation of pathogenic microorganisms including *Candida* spp., hence the susceptibility for vulvovaginal candidiasis [4,5,6]. The same effect was observed in rodents injected with oestrogen [6,7].

The incidence of vulvovaginal candidiasis during pregnancy is about 15% [8]. However, developing a *Candida* chorioamnionitis following vulvovaginal candidiasis is relatively uncommon. In a case series by Maki et al. (2017), they reported 0.3% of the deliveries had *Candida* chorioamnionitis [9]. The possible routes of infection include ascending infection from lower genital tract following premature rupture of membrane, the presence of a retained foreign body such as intrauterine contraceptive device or cervical cerclage, accidental introduction of contaminated material during amniocentesis, in vitro fertilisation or chorionic sampling, haematogenous, retrograde seeding from peritoneal cavity via fallopian tube, and systemic condition including diabetes mellitus [9,10,11,12,13,14,15].

Both our patients had gestational diabetes mellitus. Similarly, Obermair et al. (2020) reported a case of *Candida* chorioamnionitis complicated with late stillbirth in a mother with gestational diabetes mellitus, without prior obstetrics procedures [13]. It is postulated that cord vessels occlusion following infection/inflammation could be the actual cause of foetal death. In addition, there were a few other reported cases of *Candida* chorioamnionitis that were associated with diabetes mellitus, which eventually resulted in adverse perinatal outcomes [9,14,15]. As a corollary, the contributory role of diabetes mellitus in the acquisition of *Candida* chorioamnionitis following vulvovaginal candidiasis cannot be overemphasised.

As with patients with other types of diabetes mellitus, it is generally believed that pregnant women with gestational diabetes mellitus are more prone to *Candida* vaginal infection [4,5,16]. Many studies reported a high prevalence of symptomatic *Candida* infection in pregnant women who are diabetics, although the evidence was inconsistent [17,18]. Routine prophylactic antimicrobials may play a plausible role in preventing candidiasis in uncontrolled diabetics [5,19].

There are few mechanisms explaining the higher *Candida* spp. prevalence among the diabetic patients. It is widely recognised that poor glycaemic control with higher glucose serum levels aggravates yeast growth. Adherence of *Candida* spp. to vaginal surface epithelial cells is the key initiating step in yeast colonisation and subsequent infection [5,6]. Mikamo et al. (2018) revealed the increased in ICAM-1 expression on vaginal epithelial cells, which facilitates adhesion of *Candida* spp. following episodes of hyperglycaemia [20]. Intriguingly, *Candida* spp. isolated from diabetic patients was shown to be more pathogenic with higher haemolytic and esterase enzymatic activity [5]. Host factors such as the altered leukocyte function in diabetic patients together with the suppressed immune system during pregnancy contribute to recurrent vulvovaginal candidiasis [19]. In addition, vaginal secretions that contained glucose can be used as a food source by *Candida* spp. [5].

*Candida* chorioamnionitis could lead to various complications including preterm delivery, stillbirth, neonatal jaundice, sepsis, cutaneous candidiasis, risk of neurological impairment and neonatal death [9,13,14,15]. In addition, generalised *Candida* infection in a neonate may predispose to nipple thrush in the event of breast feeding, leading to premature cessation of breast feeding. The infant in one of our cases developed neonatal jaundice that eventually recovered following phototherapy. Both of them recovered well after treatment. *Candida albicans* was found to be the most common cause of *Candida* chorioamnionitis, followed by *Candida glabrata*, *Candida tropicalis*, *Candida lusitaniae*, *Candida parapsilosis*, *Candida famata* and *Candida kefyr* [9]. Relative immunosuppression of the maternal immune system such as gestational diabetes mellitus has also been found to be associated with *Candida glabrata* infection [21].

Umbilical cord examination is a crucial part in the placenta examination. Various lesions can be encountered, and some are associated with increased risk of morbidity and mortality [22]. Peripheral funisitis, the formation of umbilical cord microabscesses on its surface, is virtually pathognomonic for *Candida* infection [23,24]. Hood and colleagues were the first to describe a series of 23 cases of *Candida* chorioamnionitis. They found discrete yellowish plaques of 0.5 to 2.0 mm on the surface of umbilical cord in all cases, especially near the cord insertion site, and foci of subamniotic inflammatory exudate microscopically or also known as peripheral funisitis. Examination of the placenta showed inflammation of the foetal membrane in most instances, while villi and decidua were largely unremarkable [23].

## 4. Conclusions

*Candida* chorioamnionitis is uncommon. It has classic gross and histopathological findings on the umbilical cord, and could lead to adverse perinatal and neonatal outcomes. We recommend careful examination of the umbilical cord in mothers with gestational diabetes mellitus.

## Figures and Tables

**Figure 1 ijerph-18-07450-f001:**
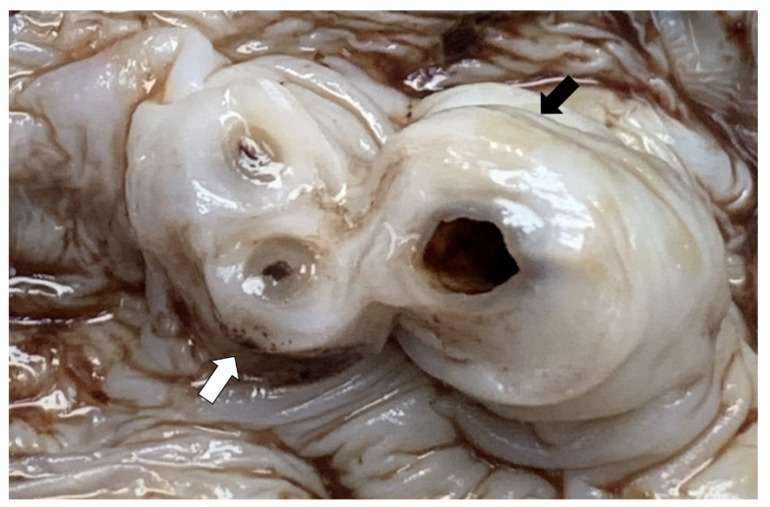
The umbilical cord demonstrates yellow (black arrow) and haemorrhagic (white arrow) spots on the surface.

**Figure 2 ijerph-18-07450-f002:**
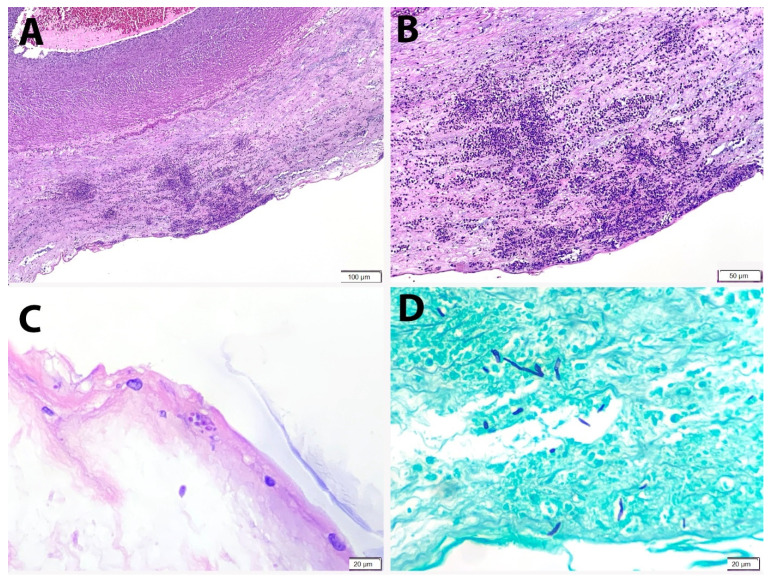
Peripheral funisitis consists of microabscesses at the subamniotic region of the umbilical cord. (**A**) H&E, 100× and (**B**) H&E, 200×. (**C**) Fungal yeasts are identified near the surface of the cord (PAS × 400). (**D**) Fungal pseudohyphae are highlighted by GMS histochemical stain (GMS × 400).

## Data Availability

Not applicable.

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
