# Peer review of "Candida Chorioamnionitis in Mothers with Gestational Diabetes Mellitus: A Report of Two Cases"

_ijerph, 2021, doi:10.3390/ijerph18147450_

Round 1

Reviewer 1 Report

The manuscript is overall well written and the cases widely documented. However, Candida infection in pregnancy and the association with gestational diabetes is not completely new. I would suggest the authors strengthen this association with more plausible references. For example, in pregnancy, how is the incidence of Candida infection in women with constitutional diabetes? 

Author Response

Reviewer 1

The manuscript is overall well written and the cases widely documented. However, Candida infection in pregnancy and the association with gestational diabetes is not completely new. I would suggest the authors strengthen this association with more plausible references. For example, in pregnancy, how is the incidence of Candida infection in women with constitutional diabetes? 

Our Responses:

Thank you for the comments. The following paragraph was added in response to the comment.

The incidence of vulvovaginal candidiasis during pregnancy is about 15%. However, developing a Candida chorioamnionitis following vulvovaginal candidiasis is relatively uncommon. In a case series by Maki et al. (2017), they reported 0.3% of the deliveries had Candida chorioamnionitis (Page 4, lines 120-123).

In addition, there were a few other reported cases of Candida chorioamnionitis that were associated with diabetes mellitus, which eventually resulted in adverse perinatal outcomes [Maki et al. 2017, Canpolat et al. 2011, Friebe-Hoffmann et al. 2000]. As a corollary, the contributory role of diabetes mellitus in the acquisition of Candida chorioamnionitis following vulvovaginal candidiasis cannot be overemphasized (Page 4, lines 133 – 137).

References below were added (See page 6, ref. 9, 14, 15)

  • Maki, Y.; Fujisaki, M.; Sato, Y.; Sameshima, H. Candidachorioamnionitis leads to preterm birth and adverse fetal-neonatal outcome. Infect Dis Obstet Gynecol., 2017, 2017, 9060138.
  • Canpolat, F.E.; Çekmez, F.; Tezer, H. Chorioamnionitis and neonatal sepsis due to Candida tropicalis. Arch Gynecol Obstet., 2011, 283(4):919-20. 
  • Friebe-Hoffmann, U.; Bender, D.P.; Sims, C.J; Rauk, P.N. Candida albicans chorioamnionitis associated with preterm labor and sudden intrauterine demise of one twin. A case report. J Reprod Med., 2000, 45(4):354-6.

Reviewer 2 Report

This MS reports 2 cases of Candida chorioamnionitis with relevant clinical findings (multiple yellowish and red spots of the surface of the umbilical cord - microabscesses with evidence of fungal yeasts and pseudohyphae). These cases were linked to a co-morbility: diabetes.

The topic of this Ms is scientifically relevant. In the end, the work has a recommendation of a careful examination of the umbilical cord of mothers with gestational diabetes.

The work is scientifically important since diabetes and fungal infections are highly increasing in the last years.

The major concern of this reviewer is that the discussion needs to be deeper that it is presented. In some cases, the MS only cites other works and does not even discuss and compares with this one. Also the scientific writing needs to be carefully reviewed.

Introduction;

  • A bit more information and references are needed here. These very recent works should be considered for this section and the discussion: https://www.ncbi.nlm.nih.gov/pmc/articles/PMC6352194/; https://pubmed.ncbi.nlm.nih.gov/33758703/; https://www.ncbi.nlm.nih.gov/pmc/articles/PMC8100220/; https://www.ncbi.nlm.nih.gov/pmc/articles/PMC8076178/;

Cassde report:

  • Line 42, 61, 63, 104: “Candida” needs to be in italic form;
  • Choose to use “Candida spp.”, “Candida sp.”, “Candida” species, or just the genera “Candida”, not all forms. Check the entire MS and uniformize this;
  • Line 63: “Streptococcus” needs to be in italic form;
  • Figure 2: measure bar needs to be indicated in the figures;
  • Line 35, 76: “gravida”?;
  • Line 134-135: “ glabrata, C. tropicalis, C. lusitaniae, C. parapsilosis, C. famata and C. kefyr [9].” – These species have not been cited before, hence, please do not use the abbreviation form;
  • Line 125: “in vitro” – lacks italic form;
  • “Obermair et al. (2020) also reported a case of Candida chorioamnionitis in a mother with gestational diabetes mellitus [10].” – Results of this study? What do the authors of this study concluded?

Author Response

Reviewer 2

This MS reports 2 cases of Candida chorioamnionitis with relevant clinical findings (multiple yellowish and red spots of the surface of the umbilical cord - microabscesses with evidence of fungal yeasts and pseudohyphae). These cases were linked to a co-morbility: diabetes. The topic of this Ms is scientifically relevant. In the end, the work has a recommendation of a careful examination of the umbilical cord of mothers with gestational diabetes. The work is scientifically important since diabetes and fungal infections are highly increasing in the last years.

The major concern of this reviewer is that the discussion needs to be deeper that it is presented. In some cases, the MS only cites other works and does not even discuss and compares with this one. Also the scientific writing needs to be carefully reviewed.

Our Response:

Thank you for the comment. We have added a few paragraphs to discuss in depth the association between diabetes in pregnancy and Candida infection, as well as the comparison between other reported cases with the current cases. (See page 4, lines 110 – 113, lines 138 – 144, lines 147 – 155).

Introduction;

  • A bit more information and references are needed here. These very recent works should be considered for this section and the discussion: https://www.ncbi.nlm.nih.gov/pmc/articles/PMC6352194/; https://pubmed.ncbi.nlm.nih.gov/33758703/; https://www.ncbi.nlm.nih.gov/pmc/articles/PMC8100220/; https://www.ncbi.nlm.nih.gov/pmc/articles/PMC8076178/;

Our Responses:

Thank you for the articles. These articles are relevant to the write-up. All the articles were reviewed, and the following sentences were added to the manuscript. During our reading, additional relevant articles that we came across were also added.

Like patients with other types of diabetes mellitus, it is generally believed that pregnant women with gestational diabetes mellitus are more prone to Candida vaginal infection [Rodrigues et al. 2019]. Many studies reported a high prevalence of symptomatic Candida infection in pregnant women who are diabetics, although the evidence was inconsistent [Nowakowska et al. 2004, Goswami et al. 2000]. Routine prophylactic antimicrobials may play a plausible role to prevent candidiasis in uncontrolled diabetics [Rodrigues et al. 2019, Mohammed et al. 2021]. (See page 4, lines 138 – 144).

Adherence of Candida spp. to vaginal surface epithelial cells is the key initiating step in yeast colonisation and subsequent infection [Rodrigues et al. 2019, d’Enfert et al. 2021]. Mikamo et al. (2018) revealed the increased in ICAM-1 expression on vaginal epithelial cells, which facilitates adhesion of Candida spp. following episodes of hyperglycaemia. Intriguingly, Candida spp. isolated from diabetic patients was shown to be more pathogenic with higher haemolytic and esterase enzymatic activity [Rodrigues et al. 2019]. Host factors such as the altered leukocyte function in diabetic patients together with the suppressed immune system during pregnancy contribute to recurrent vulvovaginal candidiasis [Mohammed et al. 2021]. Besides, vaginal secretions that contained glucose can be used as food source by Candida spp. [Rodrigues et al. 2019] (See page 4, lines 147 – 155).

References below were added (See page 6, ref. 5, 6, 17, 18, 19, 21)

  • Rodrigues CF, Rodrigues ME, Henriques M. Candida sp. Infections in Patients with Diabetes Mellitus. J Clin Med. 2019 Jan 10;8(1):76. doi: 10.3390/jcm8010076. PMID: 30634716; PMCID: PMC6352194.
  • d'Enfert C, Kaune AK, Alaban LR, Chakraborty S, Cole N, Delavy M, Kosmala D, Marsaux B, Fróis-Martins R, Morelli M, Rosati D, Valentine M, Xie Z, Emritloll Y, Warn PA, Bequet F, Bougnoux ME, Bornes S, Gresnigt MS, Hube B, Jacobsen ID, Legrand M, Leibundgut-Landmann S, Manichanh C, Munro CA, Netea MG, Queiroz K, Roget K, Thomas V, Thoral C, Van den Abbeele P, Walker AW, Brown AJP. The impact of the Fungus-Host-Microbiota interplay upon Candida albicans infections: current knowledge and new perspectives. FEMS Microbiol Rev. 2021 May 5;45(3):fuaa060. doi: 10.1093/femsre/fuaa060. PMID: 33232448; PMCID: PMC8100220.
  • Nowakowska, D.; Kurnatowska, A.; Stray-Pedersen, B.; Wilczynski, J. Prevalence of fungi in the vagina, rectum and oral cavity in pregnant diabetic women: relation to gestational age and symptoms. Acta Obstet Gynecol Scand., 2004, 83:251–
  • Goswami, R.; Dadhwal, V.; Tejaswi, S.; Datta, K.; Paul, A.; Haricharan, R.N.; Banerjee, U.; Kochupillai, N.P. Species-specific prevalence of vaginal candidiasis among patients with diabetes mellitus and its relation to their glycaemic status. Infect. 2000, 41, 162–166.
  • Mohammed L, Jha G, Malasevskaia I, Goud HK, Hassan A. The Interplay Between Sugar and Yeast Infections: Do Diabetics Have a Greater Predisposition to Develop Oral and Vulvovaginal Candidiasis? Cureus. 2021 Feb 18;13(2):e13407. 
  • Mikamo, H.; Yamagishi, Y.; Sugiyama, H.; Sadakata, H.; Miyazaki, S.; Sano, T.; Tomita, T. High glucose-mediated overexpression of ICAM-1 in human vaginal epithelial cells increases adhesion of Candida albicans. J Obstet Gynaecol., 2018, 38(2), 226-230.

Case report:

  • Line 42, 61, 63, 104: “Candida” needs to be in italic form;
  • Choose to use “Candida spp.”, “Candida sp.”, “Candida” species, or just the genera “Candida”, not all forms. Check the entire MS and uniformize this;
  • Line 63: “Streptococcus” needs to be in italic form;
  • Figure 2: measure bar needs to be indicated in the figures;
  • Line 35, 76: “gravida”?; Gravida – means number of pregnancy
  • Line 134-135: “glabrata, C. tropicalis, C. lusitaniae, C. parapsilosis, C. famata and C. kefyr [9].” – These species have not been cited before, hence, please do not use the abbreviation form;
  • Line 125: “in vitro” – lacks italic form;

Our Response: Thank you for pointing out the mistakes. All the above had been corrected as suggested.

  • “Obermair et al. (2020) also reported a case of Candida chorioamnionitis in a mother with gestational diabetes mellitus [10].” – Results of this study? What do the authors of this study concluded?

Our response:  Thank you for the comments. The results and conclusion of the study were added as suggested. (See page 4, lines 129 – 133).

Round 2

Reviewer 2 Report

Dear Authors,

Thank you for the adjustments.

The discussion is much better now.

Regards

This manuscript is a resubmission of an earlier submission. The following is a list of the peer review reports and author responses from that submission.